# COCO-Urdu: A Large Scale Urdu Image–Caption Dataset with Multimodal Quality Estimation

## Abstract

Urdu, spoken by over 250 million people, remains critically under-served in multimodal and vision-language research Joshi et al. (2020). The absence of large scale, high quality datasets has limited the development of Urdu-capable systems and reinforced biases in multilingual vision-language models trained primarily on high resource languages Bugliarello et al. (2022). To address this gap, we present **COCO-Urdu**, a large scale image-caption dataset derived from MS COCO Lin et al. (2014), containing 59,000 images and 319,000 Urdu captions selected through stratified sampling to preserve the original distribution. Captions were translated using SeamlessM4T v2 Costa-jussà et al. (2023) and validated with a hybrid multimodal quality estimation (QE) framework that integrates COMET-Kiwi for translation quality, CLIP-based similarity for visual grounding, and BERTScore with back-translation for semantic consistency; low scoring captions were iteratively refined using open-source LLMs Touvron et al. (2023). We further benchmark COCO-Urdu on BLEU Papineni et al. (2002), SacreBLEU Post (2018), and chrF Popović (2015), reporting consistently strong results. To the best of our knowledge, COCO-Urdu is the largest publicly available Urdu captioning dataset, and by releasing both the dataset and QE pipeline, we aim to reduce language bias in multimodal research and establish a foundation for inclusive vision-language systems.

## 1 Introduction

Multimodal systems that jointly reason over vision and language have advanced rapidly in recent years Radford et al. (2021); Alayrac et al. (2022). However, these gains have been disproportionately concentrated in high-resource languages such as English and Chinese, leaving low-resource communities systematically under-served Joshi et al. (2020); Ruder (2020). Urdu, spoken by over 250 million people worldwide, exemplifies this disparity: despite its large speaker base, it lacks large-scale, curated multimodal datasets that could enable high-quality captioning, retrieval, and grounded generation. The absence of such resources not only hinders Urdu-specific applications but also contributes to cross-lingual biases in multilingual models Bugliarello et al. (2022).

Prior work on Urdu image captioning remains limited. The UICD dataset Muzaffar et al. (2025) extends Flickr30k to Urdu with ∼31K images and 159K captions, but relies mainly on linguistic evaluation without systematic multimodal validation. Earlier Flickr8k-based efforts are even smaller (∼700 images) and benchmarked only with BLEU Ilahi et al. (2020). In contrast, COCO-Urdu provides substantially larger coverage and introduces scalable validation that jointly considers both semantic and visual fidelity.

In this work, we present **COCO-Urdu**, the largest Urdu image–caption dataset to date, created by translating and validating a balanced subset of MS COCO Lin et al. (2014). Our pipeline integrates complementary automatic evaluation signals, including translation quality estimation, cross-modal similarity, and semantic back-translation, to enforce alignment between images and captions at scale.

Our contributions are threefold: (i) a stratified 59K/319K Urdu caption corpus derived from MS COCO, preserving the original class distribution; (ii) a multimodal quality estimation pipeline

that enables scalable, systematic validation; and (iii) benchmarking results showing that COCO-Urdu achieves high translation quality and grounding accuracy. By releasing both the dataset and the accompanying pipeline, we aim to advance multimodal research in low-resource settings and promote more inclusive vision–language systems.

## 2 RELATED WORK

### 2.1 ZERO-SHOT TRANSLATION AND REFERENCE-FREE QUALITY ESTIMATION

Recent advances in zero-shot machine translation have enabled high-quality translation into low-resource languages without the need for parallel corpora. Models such as NLLB Fan et al. (2022) and SeamlessM4T v2 Costa-jussà et al. (2023) exemplify this paradigm, supporting translation across dozens of languages including under-represented ones. Complementary to translation, reference-free quality estimation techniques such as COMET-Kiwi Rei et al. (2020) and related models provide scalable evaluation signals, allowing large-scale pipelines to maintain semantic fidelity without relying on gold-standard references. Together, these approaches underpin modern efforts to generate multilingual datasets efficiently while controlling for quality, a strategy directly adopted in COCO-Urdu.

### 2.2 MULTIMODAL QUALITY ESTIMATION

Quality estimation (QE) for machine translation has recently been extended to multimodal settings, where images are used alongside text to assess adequacy and fidelity. Early work explored text–visual QE with transformer-based models Specia et al. (2020), while more recent efforts integrate CLIP for cross-modal alignment. CLIPScore and related variants have proven competitive for evaluating multilingual image captioning Hessel et al. (2021); Wu et al. (2024). Approaches such as CLIPTrans Yang et al. (2023) and bilingual–visual consistency models Li et al. (2024) further demonstrate that visual grounding can enhance both translation and evaluation. These findings suggest that CLIP-based QE offers a scalable and robust baseline, particularly relevant for low-resource contexts such as Urdu.

### 2.3 CLIP AND MULTIMODAL MODELS

CLIP Radford et al. (2021) has been a cornerstone of vision–language learning, enabling zero-shot transfer across tasks. Follow-up work has examined its training corpus Schuhmann et al. (2022), biases Yuksekgonul et al. (2022), and efficiency optimizations Zeng et al. (2024). While CLIP inherits limitations from its English-centric training, its strength in cross-modal alignment makes it valuable for evaluation. In COCO-Urdu, we repurpose CLIP within a custom reward-based validation pipeline (detailed in Quality Estimation Techniques Section), using it to assess image–caption consistency rather than generate captions. This shift reduces bias propagation and improves robustness in Urdu caption validation.

### 2.4 URDU IMAGE CAPTIONING

Urdu captioning datasets remain scarce and small in scale. Early efforts extended Flickr8k to Urdu with only ∼700 images and BLEU-based evaluation Ilahi et al. (2020). More recent work introduced UICD, a Flickr30k-based dataset with ∼31K images and 159K captions Muzaffar et al. (2025), but validation was primarily linguistic rather than multimodal. Other translation-based attempts have used attention-based models Ahmad et al. (2023); Ahmed et al. (2024), yet none provide systematic multimodal quality control. These limitations underscore the need for larger resources with stronger alignment guarantees.

### 2.5 LOW-RESOURCE CHALLENGES

Scaling multilingual vision–language models to low-resource languages often leads to degraded performance due to limited data and translation artifacts. Studies show that direct translation from high-resource corpora introduces semantic drifts and polarity shifts Ramesh et al. (2021), while overfitting and bias are exacerbated at small scales Kaplan et al. (2023). Recent analyses confirm that multilingual models systematically underperform on low resource languages despite strong overall

capacity Joshi et al. (2020); Ruder (2020). These findings underscore the risks of relying solely on raw machine translations. In contrast, COCO-Urdu incorporates a hybrid multimodal QE pipeline designed to detect and correct such errors, ensuring that translated captions remain both semantically faithful and visually grounded.

# 3 METHODOLOGY

## 3.1 DATASET SUBSET SELECTION

Due to computational constraints, we limited our translation efforts to a 50% subset of MS COCO. A naive random sampling approach risks introducing *class imbalance*, which can skew downstream models and impair generalization. Prior work has shown that imbalanced distributions in multi-label datasets can lead to biased representations and degraded performance Johnson & Khoshgoftaar (2019).

To mitigate this, we employed a *stratified sampling strategy*, ensuring that the relative class distributions in the subset mirror those of the full dataset. Specifically, we adapted the iterative stratification algorithm proposed by Sechidis et al. Sechidis et al. (2011), which is designed for multi-label data. This approach preserves label co-occurrence patterns while maintaining proportional representation across categories.

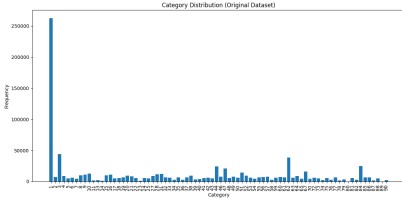

|  (a) Full MS COCO class distribution | (b) Stratified 50% subset distribution |

Figure 1: Comparison of class distributions before and after subset selection. Stratification preserves the relative frequency and co-occurrence patterns of classes, reducing risks of imbalance and skew.

## 3.2 ZERO-SHOT CAPTION TRANSLATION

To obtain Urdu captions, we employed a machine translation setting using the SeamlessM4T v2 model Costa-jussà et al. (2023), a state-of-the-art multilingual and multimodal translation system. All captions from the stratified MS COCO subset were translated into Urdu in a zero-shot manner, without the need for parallel Urdu training data. This approach leverages the model's cross-lingual generalization capabilities to extend caption coverage to a low-resource language.

## 3.3 QUALITY ESTIMATION TECHNIQUES

To ensure high-fidelity Urdu translations, we employ a hybrid ensemble of quality estimation (QE) techniques that combine NLP-based and vision-based approaches. This strategy enables scalable evaluation without relying on gold-standard references, which is crucial given the dataset size of over 319K captions.

### 3.3.1 COMET-KIWI FOR REFERENCE-FREE TRANSLATION QUALITY

We first evaluate the semantic accuracy of zero-shot translated captions using COMET-Kiwi Rei et al. (2020). Empirically, we set a threshold of 0.7 to flag low-scoring captions for iterative refinement. Across the dataset, translations achieve a mean COMET-Kiwi score of 0.76, indicating generally high semantic fidelity under a conservative threshold (Table 1).

### 3.3.2 BERTSCORE WITH BACK-TRANSLATION

To further ensure semantic consistency, we perform back-translation of Urdu captions using SeamlessM4T v2 Costa-jussà et al. (2023) and compute BERTScore Zhang et al. (2020). This reference-free approach allows large-scale comparison of semantic content and has been shown effective in multilingual MT evaluation Artetxe & Schwenk (2019). Our pipeline achieves a mean BERTScore of 0.97, suggesting that translations largely preserve the semantic content of the original captions, approaching human-level consistency.

### 3.3.3 CLIP-BASED VISUAL GROUNDING

Traditional QE methods, such as COMET or BERTScore, evaluate linguistic fidelity but ignore visual context Rohrbach et al. (2015); Liu et al. (2017). To capture cross-modal consistency, we compute a CLIP-based visual grounding score Radford et al. (2021), leveraging back-translated captions as English proxies. Let $I$ denote the CLIP image embedding, $T_{\mathrm{orig}}$ the original English caption embedding, and $T_{\mathrm{bt}}$ the back-translated caption embedding. We define

$$s_{\mathrm{orig}} = \cos(I, T_{\mathrm{orig}}), \quad s_{\mathrm{bt}} = \cos(I, T_{\mathrm{bt}})$$

and compute a relative alignment score as:

$$\mathrm{CLIPScore} = \min\left(1, 2.5 \cdot \max(s_{\mathrm{bt}}, 0) \cdot H(1, s_{\mathrm{bt}}/\max(s_{\mathrm{orig}}, \epsilon))\right)$$

where $H(\cdot)$ denotes the harmonic mean and $\epsilon$ prevents division by zero. This relative scoring accounts for the alignment quality of the original caption, rewarding translations that maintain or improve visual-text alignment and penalizing degraded captions. By integrating cross-modal signals, our pipeline captures visual-text consistency and mitigates bias propagation from imperfect source captions Chowdhery et al. (2022); Ilharco et al. (2021).

**Rationale:** Visual alignment of a translated caption depends on the quality of the source English caption. Poorly aligned source captions can make semantically correct translations appear misaligned. Our relative scoring formulation addresses this by rewarding translations that maintain or improve alignment and penalizing degraded captions. The harmonic mean term further ensures robustness by adjusting the reward based on relative improvement or decline. This design yields an interpretable, reliable metric for multilingual caption quality estimation, especially for low-resource languages like Urdu.

### 3.3.4 ENSEMBLE HYBRID SCORE

Finally, we combine COMET-Kiwi, BERTScore, and CLIP-based visual grounding into a single hybrid score for each caption. Scores are first normalized to $[0, 1]$ for comparability. The hybrid score is computed as a weighted average:

$$\mathrm{HybridScore}_i = \sum_{k \in \{\mathrm{COMET, BERT, CLIP}\}} w_k \cdot s_{i,k}, \quad \text{with} \quad \sum_k w_k = 1$$

Here, $s_{i,k}$ is the normalized score of the $i$-th caption for component $k$, and $w_k$ is the weight of that component. For COCO-Urdu, we empirically set $w_{\mathrm{COMET}} = 0.4$, $w_{\mathrm{BERT}} = 0.4$, and $w_{\mathrm{CLIP}} = 0.2$, reflecting the higher reliability of semantic evaluation relative to visual grounding.

The hybrid score systematically identifies low-quality captions for iterative refinement, leveraging complementary strengths of semantic and cross-modal evaluation.

## 4 ITERATIVE REFINEMENT OF LOW-SCORING CAPTIONS

Captions identified by the hybrid multimodal quality estimation (QE) pipeline as low-quality (QE score $< 0.7$) were subjected to an iterative refinement process. A total of 3,572 captions were automatically refined using the Qwen 14B Bai et al. (2023) language model on an NVIDIA RTX

Table 1: Quality Estimation (QE) results for COCO-Urdu captions. The ensemble of COMET-Kiwi, BERTScore, and CLIP-based visual grounding provides robust evaluation of semantic and visual fidelity.

| QE Component | Mean Score | Threshold |
|---|---|---|
| COMET-Kiwi (reference-free) | 0.76 | 0.70 |
| BERTScore with back-translation | 0.97 | 0.90 |
| CLIP-based Visual Grounding | 0.75 | 0.70 |
| Final ensemble hybrid score | 0.84 | 0.70 |

Table 2: Evaluation of low-scoring captions before and after iterative refinement.

| Metric | Before Refinement | After Refinement |
|---|---|---|
| BLEU | 0.3082 | 0.7598 |
| CHRF | 57.96 | 84.78 |
| SACREBLEU | 30.82 | 75.98 |

5090 GPU, while an additional 200 captions underwent manual correction in cases where automated refinement was insufficient.

The refinement process focused on improving sentence formulation and linguistic fluency while preserving the semantic content of the captions. This ensured that the original meaning of the translations was maintained while enhancing readability and overall linguistic quality. The observed improvements indicate that the hybrid QE pipeline effectively identified captions that scored very low on standard evaluation metrics, enabling targeted and effective refinement.

### 4.1 CAPTION-LEVEL EVALUATION

The low-scoring captions were evaluated before and after refinement using standard metrics: BLEU Papineni et al. (2002), SacreBLEU Post (2018), and CHRF Popović (2015). The results are summarized in Table 2.

The results show substantial improvements across all metrics, confirming that targeted refinement significantly enhances the quality of captions flagged as low-scoring by the hybrid QE pipeline.

Although these refined captions constitute only approximately 1% of the COCO-Urdu dataset, their improvement led to measurable gains in overall translation quality, as shown in Table 3. This demonstrates that QE-guided iterative refinement can positively impact aggregate dataset performance even when applied to a small subset of captions.

### 4.2 QUALITATIVE ANALYSIS

## 5 RESULTS

We evaluated the COCO-Urdu captions using both reference-free and reference-based metrics. Reference-free evaluation was performed using our ensemble quality estimation (QE) pipeline, which integrates COMET-Kiwi, BERTScore, and CLIP-based visual grounding (see Table 1). These metrics guided iterative refinement of low-quality captions, resulting in a high-quality final dataset.

For reference-based evaluation, we computed standard machine translation (MT) metrics, including BLEU Papineni et al. (2002), SacreBLEU Post (2018), and CHRF Popović (2015). Human reference translations are unavailable at this scale, so we generated reference translations using the NLLB-3B model Costa-Jussà et al. (2022), which has been shown to achieve near-human quality for low-resource languages. This approach allows reliable automated evaluation of large-scale datasets. Zero-shot translations were obtained using SeamlessM4T Costa-jussà et al. (2023), which were subsequently refined via the QE pipeline.

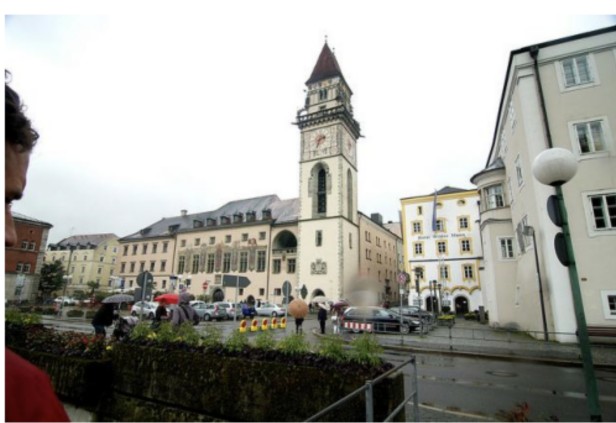

original machine transalation: اس میں کاروں کے ساتھ ایک شہر پارکنگ بہت سے کی ایک تصویر
refined machine transalation: ایک شہر کی پارکنگ کی تصویر جس میں کاریں ہیں۔

Figure 2: Representative examples of captions before and after iterative refinement, demonstrating improved sentence formulation and fluency while preserving the original meaning.

Table 3: Reference-based MT evaluation of COCO-Urdu captions and other high-performing Urdu image captioning datasets. Reference translations for COCO-Urdu were generated using NLLB-3B Costa-Jussà et al. (2022).

| Dataset | Images/Captions | BLEU | SacreBLEU | CHRF |
|---|---|---|---|---|
| COCO-Urdu (Refined) | 59K/319K | 0.53 | 53 | 74 |
| COCO-Urdu (Zero-shot) | 59K/319K | 0.52 | 52 | 73.23 |
| UICD Muzaffar et al. (2025) | 31K/135K | 0.86 | | |
| Flickr8k Urdu Ilahi et al. (2020) | 700/700 | 0.83 | | |

*Note: Despite its much larger and more diverse scale, COCO-Urdu achieves performance on par with smaller datasets. UCID's evaluation process is less documented, and Flickr8k-Urdu was limited to only 700 images from a narrow domain, which may artificially inflate BLEU scores.*

Although reference-based metrics were not directly optimized during QE-guided refinement, COCO-Urdu captions score highly, demonstrating that our ensemble approach produces translations with strong semantic fidelity and cross-modal alignment.

## 5.1 QUANTITATIVE RESULTS

Table 3 presents a comparison of COCO-Urdu with other high-performing Urdu image caption datasets, reporting metrics before and after QE-guided refinement.

## 5.2 DISCUSSION

The improvement from zero-shot to refined translations demonstrates the effectiveness of our ensemble QE pipeline. Despite primarily using reference-free evaluation for refinement, COCO-Urdu performs well on reference-based metrics, highlighting the high semantic fidelity and cross-modal consistency of the captions. Leveraging NLLB-3B for reference translation ensures reliable automated scoring at this scale and confirms that combining NLP and vision-based QE techniques is an effective strategy for producing and validating large-scale low-resource language datasets.

## 6   FAULT-TOLERANT PARALLEL TRANSLATION PIPELINE

To efficiently translate the COCO-Urdu subset at scale, we designed a fault-tolerant, parallelizable pipeline with the following key steps:

1. **Dataset Splitting:** The dataset is partitioned into non-overlapping ranges, creating discrete chunks for independent processing. Each range is associated with a unique version.

2. **Version Checking and Safe Retriggers:** Before translation, the pipeline checks if a given range already exists on Hugging Face. If the version is present, it is skipped, enabling safe re-execution of failed or interrupted chunks without overwriting previous results.

3. **Translation and Quality Estimation:** Each chunk undergoes zero-shot translation via SeamlessM4T, followed by our ensemble quality estimation pipeline: COMET-Kiwi, BERTScore with back-translation, and CLIP-based visual grounding.

4. **Versioned Storage:** Results for each chunk are uploaded to Hugging Face with versioning based on the range, ensuring reproducibility and traceability.

5. **Parallel Execution:** Independent chunks can be processed simultaneously across heterogeneous compute platforms (e.g., A100 40GB, RTX 5090 32GB), reducing overall runtime. While a single-GPU estimate was approximately 20 hours, parallel execution reduced translation and QE for the entire dataset to ∼4 hours.

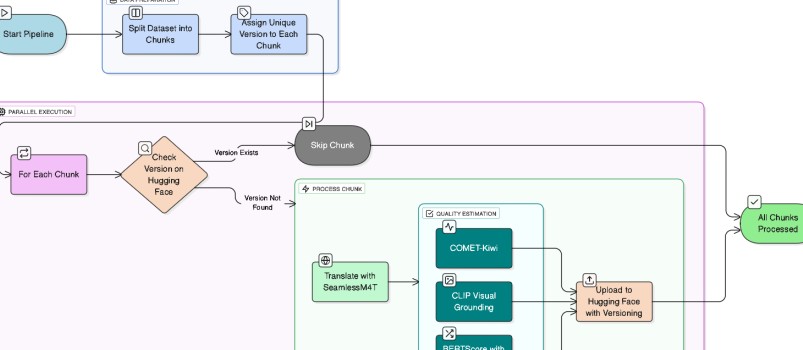

Figure 3: Schematic of the fault-tolerant, parallel COCO-Urdu translation pipeline. Each chunk is processed independently with versioned outputs and integrated QE steps.

**Advantages:** This design ensures fault tolerance, reproducibility, and efficient resource utilization. Failed or interrupted jobs can be retried safely, translation can be distributed across multiple devices, and overall runtime scales linearly with available compute resources.

## 7   HUMAN EVALUATION SCOPE AND LIMITATIONS

Human evaluation in this work was intentionally restricted to 200 captions. This decision was shaped by both methodological and practical considerations.

Methodologically, the central aim was to test the effectiveness of the proposed hybrid quality estimation (QE) framework. Human judgments were therefore used in a targeted manner, primarily to validate low-scoring captions flagged by the QE pipeline. This design choice highlights the potential of QE to reduce dependence on exhaustive manual annotation while maintaining dataset quality.

Practically, budget and time constraints limited the feasibility of large-scale crowdsourced evaluation. Within these constraints, we prioritized demonstrating the viability of QE-driven validation over comprehensive human annotation.

This trade-off inevitably leaves certain linguistic subtleties and cultural nuances underexplored, and broader human validation will be necessary before deploying the dataset in high-stakes downstream tasks. Nevertheless, the current scope establishes a basis for future work in expanding human evaluation, refining captions, and experimenting with alternative QE-guided annotation strategies.

## 8 DISCUSSION AND FUTURE WORK

COCO-Urdu advances multimodal research for low-resource languages by contributing both a large-scale Urdu image caption dataset and a systematic hybrid QE framework. By integrating semantic and visual signals into a single score, our approach moves beyond traditional translation metrics and provides a scalable method for dataset validation that is both interpretable and inclusive.

Building on this foundation, future research may extend COCO-Urdu in several directions. First, human evaluation can be expanded to capture a wider range of linguistic and cultural variations, complementing the automatic QE pipeline. Second, the dataset enables the training and fine-tuning of Urdu-specific vision language models for tasks such as captioning, retrieval, and multimodal reasoning. Third, adapting large multilingual vision models to Urdu and benchmarking their zero-shot performance represents a promising avenue. Finally, the dataset has potential utility in downstream applications, including educational technologies, assistive systems, and inclusive content generation.

More broadly, the methodology outlined here, combining hybrid QE with targeted refinement, offers a generalizable framework for other low-resource languages and contributes to the development of equitable and globally representative multimodal AI.

## 9 CONCLUSION

We introduced **COCO-Urdu**, the largest publicly documented Urdu image–caption dataset to date, alongside a hybrid multimodal quality estimation framework that integrates semantic and visual evaluation. By combining COMET-Kiwi, BERTScore with back-translation, and CLIP-based visual grounding, we demonstrated a scalable method for validating translations at scale, reducing reliance on exhaustive human annotation. Our iterative refinement of low-scoring captions further showed that targeted intervention can significantly enhance overall dataset quality.

COCO-Urdu directly addresses the lack of large-scale multimodal resources for Urdu, a language spoken by over 250 million people yet critically under-served in vision–language research. Beyond its immediate contributions, the dataset establishes a foundation for developing and fine-tuning Urdu-capable captioning, retrieval, and multimodal reasoning systems. More broadly, our methodology offers a generalizable blueprint for constructing inclusive datasets in other low-resource languages, promoting equity and reducing cross-lingual biases in multimodal AI.

### ACKNOWLEDGMENTS

Use unnumbered third level headings for the acknowledgments. All acknowledgments, including those to funding agencies, go at the end of the paper.

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

## A  APPENDIX: DATASET LICENSING

The COCO dataset Lin et al. (2014) is released under the Creative Commons Attribution 4.0 International License (CC BY 4.0), which permits sharing, adaptation, and commercial use, provided appropriate credit is given. Note that the images in COCO were sourced from Flickr and are subject to Flickr's Terms of Use; users must ensure compliance with these terms when utilizing the images.

The COCO-Urdu dataset presented in this work is a derivative of the original COCO dataset, consisting of translated captions into Urdu. In accordance with licensing requirements for derivative works:

- The original COCO license is retained, and proper attribution is given.
- Modifications made to the dataset are clearly described (i.e., translation of captions into Urdu).
- The translated captions are released under CC BY 4.0, allowing others to use, share, and adapt the modifications while providing appropriate attribution.

This ensures that both the original dataset and the modifications are properly licensed, promoting responsible use and enabling further research in low-resource language image captioning.

