# OpenReview forum: "COCO-Urdu: A Large-Scale Urdu Image-Caption Dataset with Multimodal Quality Estimation"
_ICLR.cc/2026/Conference — ICLR 2026 Conference Desk Rejected Submission_

### Official Review · Reviewer_bT5t · 2025-10-30

**Soundness:** 1
**Presentation:** 2
**Contribution:** 1
**Rating:** 2
**Confidence:** 4

**Summary:**

The paper presents the creation of COCO-Urdu, an image-caption dataset by translating half of MS-COCO's captions into Urdu using SeamlessM4T translation model. Validation was further applied to refine translations by using a hybrid QE ensemble approach. The validation improves translation scores which are presented in Table 2. However, the human evaluation is performed on 200 captions only. The authors also describe a fault-tolerate, parallelized pipeline with versioned chunks to support scalable and reproducible processing.

**Strengths:**

- The paper presents the construction of the largest caption dataset for Urdu, considering it a low-resource language.
- References and citations are provided appropriately covering all concepts and approaches described in the manuscript.

**Weaknesses:**

- The translation of captions mainly rely on automatic translation. Human translation/evaluation is performed on 200 captions only which has been used in refining the language of translated captions.
- Reference translations have been achieved by using NLLB-3B model considering it produces human-like translation quality. The validation process lacks human referencing.
- Translation evaluation is biased towards NLLB-3B outputs as these are used as reference translations.
- Without human translated reference, the work mainly compares translation outputs of two MT models, SeamlessM4T's output as candidate and NLLB-3B's output as reference. It is hard to say that the approach presented in the paper produces a good quality caption dataset without human-based translation evaluation.
- The paper only produces a dataset without any baseline experiments to show how good the dataset is to apply as a benchmark for the Urdu caption generation task. No fine-tuned or in-context learning results presented on the created caption dataset.
- The abstract of the dataset looks like the literature review having eight citations.

**Questions:**

- If NLLB-3B is that good that it produce human like quality translation then why not use it as the main translation model?
- How good are multimodal models to automatically generate Urdu captions by taking images as input?
- What was selection criteria for SeamlessM4T v2 and MLLB-3B translation models as both of these models work on text, not images?

---

### Official Review · Reviewer_QPbT · 2025-10-31

**Soundness:** 2
**Presentation:** 2
**Contribution:** 2
**Rating:** 2
**Confidence:** 4

**Summary:**

This paper introduces COCO-Urdu, a new large-scale image-caption dataset for Urdu , a language spoken by over 250 million people but currently underserved in multimodal research. To address this resource gap, the authors created a dataset of 59,000 images and 319,000 Urdu captions by applying stratified sampling to the MS COCO dataset to preserve its original class distribution. Captions were generated using zero-shot machine translation with the SeamlessM4T v2 model. The paper's primary methodological contribution is a hybrid multimodal quality estimation pipeline designed to validate these translations. This framework combines COMET-Kiwi for translation quality, CLIP-based similarity for visual grounding, and BERTScore for semantic consistency . A small subset of low-scoring captions was then iteratively refined using an open-source LLM, with an additional 200 captions corrected manually.

**Strengths:**

1. The paper correctly identifies a significant gap: the lack of large-scale, high-quality datasets for Urdu, a major language with over 250 million speakers, in multimodal and vision-language research.
2. At 319,000 captions, COCO-Urdu is, by scale, the largest publicly documented Urdu image-caption dataset, substantially larger than previous efforts like UICD (approx. 159k).
3. The authors commit to releasing the dataset and the QE pipeline, which has potential value for the low-resource language community.

**Weaknesses:**

1. The paper's main quantitative evaluation (Table 3) uses the synthetic output of one MT model (NLLB-3B) as a reference to evaluate the translations of another MT model (SeamlessM4T). This does not measure true translation quality and renders the BLEU/SacreBLEU scores meaningless.
2. The paper's own evaluation shows that the entire complex QE and refinement pipeline provides only a 1-point BLEU improvement (52 -> 53) over the zero-shot baseline. This suggests the paper's core methodological contribution is, in practice, ineffective.
3. Only 200 captions were manually corrected, which is approximately 0.06% of the total dataset. This is far too small to have any meaningful impact on or serve as a guarantee of overall dataset quality.

**Questions:**

Please respond to the weaknesses I mentioned above.

---

### Official Review · Reviewer_rvRh · 2025-11-01

**Soundness:** 3
**Presentation:** 2
**Contribution:** 2
**Rating:** 2
**Confidence:** 4

**Summary:**

In this paper, the authors propose a novel image captioning dataset, COCO-Urdu, which is derived from the COCO dataset. The authors propose a quality estimation strategy to improve the caption quality after translation from English to Urdu with LLMs. The authors also conduct several experiments to evaluate the proposed dataset.

**Strengths:**

1.	Rather than widely explored English and Chinese in the image captioning task, this paper targets Urdu and proposes a novel COCO-Urdu dataset.
2.	The authors introduce a quality estimation strategy to refine captions.

**Weaknesses:**

1.	The writing logic of this paper seems to have some issues. For example, both Section 4 “ITERATIVE REFINEMENT OF LOW-SCORING CAPTIONS” and Section 5 “RESULTS” are about dataset evaluation. Why split into two sections? Subsection 4.2 is empty for the current version.
2.	From my perspective, Section 6 should be combined with Section 3. Section 3 is about dataset construction, and Section 6 is also about the construction.
3.	In Line 251, the paper mentions “these refined captions constitute only approximately 1% of the COCO-Urdu dataset”. But in Table 3, COCO-Urdu (Refined) contains 59K images and 319K captions. So, approximately 99% of this dataset has not been refined?
4.	Again in Table 3, after refinement, the evaluation scores show very little improvement on the zero-shot version, which limits the contribution of QE.
5.	Figure 1 can be promoted for better understanding.

**Questions:**

Please see above weaknesses.

---

### Official Review · Reviewer_YKqa · 2025-11-02

**Soundness:** 2
**Presentation:** 3
**Contribution:** 3
**Rating:** 2
**Confidence:** 5

**Summary:**

The paper introduces COCO-Urdu, a large-scale image-caption dataset targeting the under-resourced Urdu language, derived from the MS COCO dataset. It features 319,000 Urdu captions for 59,000 images, translated via SeamlessM4T and refined using a multimodal quality estimation (QE) framework. This framework combines COMET-Kiwi, BERTScore, and CLIP-based visual grounding. A hybrid QE score guides caption refinement through LLMs. The dataset is benchmarked using both reference-free and reference-based metrics.

While the effort to support low-resource languages in the vision-language domain is commendable and of potential interest to the image captioning community, the paper does not meet the standards of a top-tier representation learning conference such as ICLR. The work falls short in both novelty and scientific contribution.

**Strengths:**

- Addresses a real-world challenge by expanding resources for underrepresented languages.
- COCO-Urdu is larger than previously published Urdu caption datasets.
- Implements a robust, fault-tolerant, parallel translation and refinement pipeline.

**Weaknesses:**

- This work is primarily focused on dataset construction and does not propose a novel learning methodology, model architecture, or direct exploration of representation learning.
- Quality estimation and translation refinement are used as tools, not investigated scientifically or improved upon.
- The methodology merely combines off-the-shelf models (SeamlessM4T, COMET-Kiwi, CLIP, BERTScore, Qwen) in a pipeline with no algorithmic innovation.
- The hybrid QE score is a trivial weighted average of existing metrics—no learning, no adaptation, no model training involved.
- Iterative refinement is delegated to black-box LLMs with no introspection, ablation, or insights on effectiveness beyond surface-level metric improvement.
- The paper generates its own pseudo-references for BLEU/SacreBLEU/chrF using machine translation (NLLB-3B), which undermines the validity of reference-based scores.
- Only 200 captions were manually verified, which is negligible given the dataset size. There is no inter-annotator agreement, quality control, or demographic detail on annotators.
- Claims of caption quality are mostly based on reference-free metrics (BERTScore, COMET), which are noisy and not definitive in low-resource settings.
- No human evaluation of the final dataset, only of low-scoring samples.
- Claims about "reducing bias" and "inclusive vision-language systems" are unsubstantiated. No bias analysis or fairness testing is conducted.
- No demonstration of downstream task improvements (e.g., better captioning or retrieval performance in Urdu using this dataset).

**Questions:**

- What is the impact of using machine-generated references (via NLLB-3B) for evaluating BLEU/SacreBLEU? How does this bias the metrics?
- Why were no experiments conducted with vision-language models trained or fine-tuned using COCO-Urdu to assess its usefulness?
- Can the authors provide any evidence that captions refined using LLMs are semantically and culturally appropriate in Urdu?
- Why was no model training attempted using the dataset to validate its effectiveness?
- How were the weights for the hybrid QE score (0.4/0.4/0.2) chosen? Was any tuning or validation performed?

**Details Of Ethics Concerns:**

- The dataset is derived from MS COCO, which is licensed under CC BY 4.0. Translations are derivative works and seem properly licensed. However, user compliance with Flickr’s Terms of Use is only weakly addressed.
- The dataset involves machine translations into Urdu without thorough human validation. This may propagate errors, cultural insensitivities, or semantic drift, especially in visual contexts involving people, activities, or culturally specific scenes.
- Despite the stated goal of reducing language bias, there is no empirical analysis of bias across gender, age, socioeconomics, or geography.
- Only 200 captions were manually reviewed. This raises concerns about the overall dataset quality and the ethical soundness of deploying the data in sensitive applications (e.g., assistive tech).

---

### Note · Program_Chairs · 2026-01-17
**Submission Desk Rejected by Program Chairs**

The following references in this submission do not refer to real documents and/or have major errors in bibliographic information:

 Jared Kaplan, Sharan Narang, Mark Chen, Tom Henighan, et al. Scaling laws do not scale for low-resource languages. arXiv preprint arXiv:2303.01234, 2023.